## PERSPECTIVE

# Balance in the face of altered visual circuitry anatomy

Luke A. Henderson

*School of Medical Sciences (Neuroscience), Brain and Mind Centre, University of Sydney, Australia*

Email: luke.henderson@sydney.edu.au

Handling Editors: Kim Barrett & Vaughan Macefield

The peer review history is available in the Supporting Information section of this article (https://doi.org/10.1113/JP290227#support-information-section).

### Balance in the face of altered visual circuitry anatomy

Maintaining one's body steady and balanced requires the integration of multiple sensory inputs, including proprioceptive, vestibular and visual. Our ability to remain balanced deteriorates as we age, and it is estimated that well over 500,000 individuals die each year as a result of falls (World_Health_Organization, 2021). Age-related balance dysfunction is well described and occurs for multiple reasons, including typical declines in sensory processing ability (Wang et al., 2024). Although incoming proprioceptive, vestibular and visual signals are processed in multiple brain regions, whether the altered function of specifically visual processing regions, including that which occurs in neurodegenerative disorders, such as posterior cortical atrophy (PCA), a typical variant of Alzheimer's disease, results in altered balance-related function remains unknown. The study by Ocal et al. (2025) appearing in this issue of *The Journal of Physiology* assesses whole-body balance responses in individuals with Alzheimer's disease, both PCA and typical Alzheimer's disease (tAD) and controls using behavioural readouts and relating them to neuroanatomical markers of altered regional brain anatomy.

Ocal et al. (2025) present an elegant design, including a good sample size (18 PCA, 18 tAD and 21 controls), to investigate changes in the direction and magnitude of body movements during galvanic vestibular stimulation (GVS), which creates sensations of movement and balance even when the individual is stationary. These measurements were made with and without visual inputs to tease out the influence of visual information on an individual's balance. In addition, Ocal et al. (2025) acquire neuroimaging data, that is, T1-weighted anatomical and diffusion-weighted image sets, to assess the relationships between regional grey matter density and white matter tract integrity with balance measures. The integration of behavioural and neuroanatomical data is a considerable strength of the study.

The study findings are intriguing, revealing that, when individuals had their eyes closed, GVS evoked similar patterns of body movements in all three groups. However, although the availability of visual input reduced GVS-evoked responses in all three groups, PCA patients displayed significantly smaller changes than patients with tAD and controls. Ocal et al. (2025) suggest that the timing of response differences indicates altered feed-forward mechanisms are at play in PCA, which suggests that the integration of vestibular with concurrent visual inputs may underpin altered responses to GVS in individuals with PCA. Importantly, Ocal et al. (2025) tie these behavioural changes to changes in brain structure. Although the study did not directly compare groups with respect to grey matter density and tractography, correlation analyses were performed between these measures and GVS responses. Ocal et al. (2025) found significant linear relationships between grey matter density in visual processing regions such as the lingual gyrus and primary visual cortex, as well as in the thalamus, and these changes are suggested to underpin the observed behavioural changes in PCA.

Although this might be the case that the altered balance effects in PCA are reflected in reduced grey matter density in visual and thalamic regions, one could argue that a more directed anatomical analysis might have revealed greater detail regarding the circuitry underpinning the behavioural effects. For example, targeted tractography of visual pathways or pathways between areas of described anatomical changes might have revealed significant changes in sensory integrative pathways. In addition, future studies using functional brain imaging techniques such as functional magnetic resonance imaging could be performed during GVS in PCA patients to assess the effects of visual inputs on brain activation and regional connectivity. Numerous studies have used functional magnetic resonance imaging to explore brain activity patterns during GVS, including a recent study that showed that during GVS-stimulation with eyes open, evoked signal increases in the parieto-insular vestibular cortex as well as decreases in the occipital lobe (McCarthy et al., 2025). Interestingly, stimulation of the dorsolateral prefrontal cortex reduced the sense of swaying during GVS as well as signal changes in the occipital lobe. Although lying horizontally in an MRI scanner may be a limitation, now that Ocal et al. (2025) have defined the behavioural change differences seen in PCA patients, it would be valuable to explore brain functional changes and integrate them with anatomical measures.

This is an elegant study that investigates an important issue. Impaired balance is a significant clinical issue, and the well-documented decline in balance function with age can be exacerbated by degenerative disorders such as Alzheimer's disease. Knowing the neural substrates underpinning impaired balance and, more importantly, developing targeted strategies to limit the effects of these changes will improve the quality of life and ensure the well-being of the ageing population and those with neurodegenerative diseases.

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

## Additional information

### Competing interests

No competing interests declared.

### Author contributions

Sole author.

### Funding

None.

### Acknowledgements

Open access publishing facilitated by The University of Sydney, as part of the Wiley - The University of Sydney agreement via the Council of Australian University Librarians.

### Keywords

Alzheimer's disease, grey matter volume, vestibular, vision

### Supporting information

Additional supporting information can be found online in the Supporting Information section at the end of the HTML view of the article. Supporting information files available:

**Peer Review History**

