## [Peer Review History · The Journal of Physiology]

Balance in the face of altered visual circuitry anatomy

Luke A Henderson

DOI: 10.1113/JP290227

Corresponding author(s): Luke Henderson (luke.henderson@sydney.edu.au)

The following individual(s) involved in review of this submission have agreed to reveal their identity: Keir Yong (Referee #1)

Review Timeline:	Submission Date:	10-Nov-2025
	Editorial Decision:	13-Nov-2025
	Revision Received:	19-Dec-2025
	Accepted:	07-Jan-2026

Senior Editor: Kim Barrett

Reviewing Editor: Vaughan Macefield

Transaction Report:

Dear Dr Henderson,

Re: JP-P-2025-290227 "**Balance in the face of altered visual circuitry anatomy**" by Luke A Henderson

Thank you for submitting your manuscript to The Journal of Physiology. It has been assessed by a Reviewing Editor and by 1 expert referee and we are pleased to tell you that it is acceptable for publication following satisfactory revision.

The review comments are copied at the end of this email.

Please address all the points raised and incorporate all requested revisions or explain in your Response to Referees why a change has not been made. We hope you will find the comments helpful and that you will be able to return your revised manuscript within 2 weeks. If you require longer than this, please contact journal staff: jp@physoc.org.

REVISION CHECKLIST:

We look forward to receiving your revised submission.

Yours sincerely,

Kim Barrett
Senior Editor
The Journal of Physiology

REQUIRED ITEMS

- 1) - The reference list must be in alphabetical order, rather than numbered, to comply with our Journal format.
- 2) - Please include a full, separate title page as part of your main article (Word) file, which should contain the following: title, authors, affiliations, corresponding author name and contact details, keywords, and running title.
- 3) - The corresponding author must provide an institutional email address (not a personal address) in the manuscript.

EDITOR COMMENTS

Reviewing Editor:

Thank you for submitting your Perspective article to The Journal of Physiology, which has been reviewed by the corresponding author of the article to which you refer. As you will see, there are some minor issues you will need to address before we can accept your manuscript. In addition to these issues, please cite the references in the article according to Journal style (Surname et al., year), remove the details of the target article from page 2 and cite it in the reference list and with the text. Finally, please include your email address and provide keywords on the first page.

REFEREE COMMENTS

Referee #1:

Thank you for this Perspective article which provides clinical context for key findings along with potential avenues for further research.

A particular area of interest is the interplay between parieto-insular and occipital cortical activation during galvanic vestibular stimulation (GVS; McCarthy et al., 2025). This may align with Ocal et al.'s Discussion (p14) on:

"...human brain activation studies have demonstrated intimate and inhibitory interactions between visual and vestibular cortical regions, including visual cortical responses to optokinetic stimulation accompanied by deactivation of vestibular cortices. These findings suggest disruption to a cortico-thalamic network inhibiting vestibular-driven balance response based on conflicting visual information."

Main points:

#1 There appears to be a typo in the sentence reporting differences in response to GVS in participants with posterior cortical atrophy (PCA), typical Alzheimer's disease (tAD) and controls.

The third paragraph suggests that group responses to GVS were comparable with eyes open. However, responses were comparable with eyes closed and with the head at different orientations, and were modulated by vision to a lesser extent in PCA relative to tAD and control groups. This is evidence of altered visuo-vestibular integration owing to PCA despite comparable proprioceptive-vestibular integration across groups.

This is particularly interesting given clear disturbances in verticality perception- whether judged visually or haptically- in the same PCA group (Day et al., JPhysiol, 2022). This suggests separability of systems responsible for spatially transforming these sensory signals - one responsible for rapid self-motion detection and motor control, the other for perceiving the world in a stable, upright manner.

I suggest amending relevant text to:

"Their findings are intriguing, revealing when individuals had their eyes closed, GVS evoked similar patterns of body movements in all three groups. However, whilst the availability of visual input reduced GVS-evoked responses in all three groups, PCA patients displayed significantly smaller changes than patients with tAD and controls."

Minor points

#2 Typos: 'variate'-> variant; 'PAC' -> PCA; 'bran' -> brain

#3 'subsequent access' vestibular (GVS polarity), proprioceptive (head facing left, right, straight ahead) and vision (without, with) conditions were randomised.

END OF COMMENTS

Dear Editor,

I thank the reviewers for their comments and have addressed each of them below

Regards,

Luke

Reviewing Editor:

1) Please cite the references in the article according to Journal style (Surname et al., year), remove the details of the target article from page 2 and cite it in the reference list and with the text. Finally, please include your email address and provide keywords on the first page.

These changes have been made.

Referee #1:

1) There appears to be a typo in the sentence reporting differences in response to GVS in participants with posterior cortical atrophy (PCA), typical Alzheimer's disease (tAD) and controls. The third paragraph suggests that group responses to GVS were comparable with eyes open. However, responses were comparable with eyes closed and with the head at different orientations, and were modulated by vision to a lesser extent in PCA relative to tAD and control groups. This is evidence of altered visuo-vestibular integration owing to PCA despite comparable proprioceptive-vestibular integration across groups. This is particularly interesting given clear disturbances in verticality perception- whether judged visually or haptically- in the same PCA group (Day et al., JPhysiol, 2022). This suggests separability of systems responsible for spatially transforming these sensory signals - one responsible for rapid self-motion detection and motor control, the other for perceiving the world in a stable, upright manner. I suggest amending relevant text to:

"Their findings are intriguing, revealing when individuals had their eyes closed, GVS evoked similar patterns of body movements in all three groups. However, whilst the availability of visual input reduced GVS-evoked responses in all three groups, PCA patients displayed significantly smaller changes than patients with tAD and controls."

My apologies – this was indeed a typo and the suggested change has been made – thank you for identifying this as it was a critical mistake.

2) Typos: 'variate' -> variant; 'PAC' -> PCA; 'bran' -> brain

These types have been corrected.

3) *'subsequent access' vestibular (GVS polarity), proprioceptive (head facing left, right, straight ahead) and vision (without, with) conditions were randomised.*

My apologies but I am unclear as to what this is referring to.

Dear Professor Henderson,

Re: JP-P-2025-290227R1 "**Balance in the face of altered visual circuitry anatomy**" by Luke A Henderson

We are pleased to tell you that your paper has been accepted for publication in The Journal of Physiology.

Please note that Perspective articles are not typically covered by institutional open access agreements with our publisher, Wiley. Wiley do not offer article processing charge (APC) discounts for smaller article types in hybrid subscription journals, meaning that if you wish for your Perspective to be published Open Access, you will have to pay the full APC. As such, we recommend authors publish Perspectives 'behind the paywall', where they will become freely accessible after a 12-month embargo (i.e. please select the NON open access option via Wiley Author services during proofing).

Should you wish to pay for Open Access, you will be able to place an order by logging into Wiley Author services.

Yours sincerely,

Kim Barrett
Senior Editor
The Journal of Physiology

IMPORTANT POINTS TO NOTE FOLLOWING ACCEPTANCE OF YOUR PAPER:

- **IMPORTANT NOTICE ABOUT OPEN ACCESS:** To assist authors whose funding agencies mandate immediate public access to published research findings, The Journal of Physiology allows authors to pay an Open Access (OA) fee to have their papers made freely available immediately on publication.

- You can help your research get the attention it deserves! Check out Wiley's free Promotion Guide for best-practice recommendations for promoting your work at: www.wileyauthors.com/eoo/guide. You can learn more about Wiley Editing Services which offers professional video, design, and writing services to create shareable video abstracts, infographics, conference posters, lay summaries, and research news stories for your research at: www.wileyauthors.com/eoo/promotion.

- If you would like to receive our 'Research Roundup', a monthly newsletter highlighting the cutting-edge research published in The Physiological Society's family of journals (The Journal of Physiology, Experimental Physiology, Physiological Reports, The Journal of Nutritional Physiology and The Journal of Precision Medicine: Health and Disease), please click this link, fill in your name and email address and select 'Research Roundup': <https://www.physoc.org/journals-and-media/membernews>

EDITOR COMMENTS

Reviewing Editor:

Thank you for submitting your revised manuscript to The Journal of Physiology. The reviewer and I are both satisfied with your responses.

REFEREE COMMENTS

Referee #1:

The authors have addressed my points.